# The endoplasmic reticulum, not the pH gradient, drives calcium refilling of lysosomes

**Abigail G Garrity[1†], Wuyang Wang[2†], Crystal MD Collier[2], Sara A Levey[2], Qiong Gao[2], Haoxing Xu[1,2]\***

[1]Neuroscience Program, University of Michigan, Ann Arbor, United States; [2]Department of Molecular, Cellular, and Developmental Biology, University of Michigan, Ann Arbor, United States

**Abstract** Impaired homeostasis of lysosomal $Ca^{2+}$ causes lysosome dysfunction and lysosomal storage diseases (LSDs), but the mechanisms by which lysosomes acquire and refill $Ca^{2+}$ are not known. We developed a physiological assay to monitor lysosomal $Ca^{2+}$ store refilling using specific activators of lysosomal $Ca^{2+}$ channels to repeatedly induce lysosomal $Ca^{2+}$ release. In contrast to the prevailing view that lysosomal acidification drives $Ca^{2+}$ into the lysosome, inhibiting the V-ATPase $H^+$ pump did not prevent $Ca^{2+}$ refilling. Instead, pharmacological depletion or chelation of Endoplasmic Reticulum (ER) $Ca^{2+}$ prevented lysosomal $Ca^{2+}$ stores from refilling. More specifically, antagonists of ER IP3 receptors (IP3Rs) rapidly and completely blocked $Ca^{2+}$ refilling of lysosomes, but not in cells lacking IP3Rs. Furthermore, reducing ER $Ca^{2+}$ or blocking IP3Rs caused a dramatic LSD-like lysosome storage phenotype. By closely apposing each other, the ER may serve as a direct and primary source of $Ca^{2+}$ for the lysosome.

**\*For correspondence:** haoxingx@ umich.edu

[†]These authors contributed equally to this work

**Competing interests:** The authors declare that no competing interests exist.

## Introduction

A vacuolar-type $H^+$-ATPase (V-ATPase) on the membrane of the lysosome maintains the acidic lumen ($pH_{Ly} \sim 4.6$), and improper acidification of lysosomes may lead to lysosomal storage diseases (LSDs) (*Mindell, 2012*). Like the Endoplasmic Reticulum (ER) (*Clapham, 2007*; *Berridge, 2012*), lysosomes are also intracellular $Ca^{2+}$ stores with free $[Ca^{2+}]_{Ly} \sim 0.4$–$0.6$ mM (*Christensen et al., 2002*; *Lloyd-Evans et al., 2008*), which is 3–4 orders of magnitude higher than the cytosolic $[Ca^{2+}]$ ($\sim 100$ nM). A reduction in $[Ca^{2+}]_{Ly}$ is believed to be the primary pathogenic cause for some LSDs and common neurodegenerative diseases (*Lloyd-Evans et al., 2008*; *Coen et al., 2012*). Using the fast $Ca^{2+}$ chelator BAPTA, $Ca^{2+}$ release from the lysosome has been shown to be required for late endosome-lysosome fusion (*Pryor et al., 2000*), lysosomal exocytosis, phagocytosis, membrane repair, and signal transduction (*Reddy et al., 2001*; *Lewis, 2007*; *Kinnear et al., 2004*). Consistently, the principal $Ca^{2+}$ channel in the lysosome, Mucolipin TRP channel 1 (TRPML1 or ML1), as well as lysosomal $Ca^{2+}$ sensors such as the C2 domain–containing synaptotagmin VII, are also required for these functions (*Steen et al., 2007*; *Lewis, 2007*; *Kinnear et al., 2004*). Whereas human mutations of *TRPML1* cause type IV Mucolipidosis, pathogenic inhibition of ML1 underlies several other LSDs (*Shen et al., 2012*).

How the 5000-fold $Ca^{2+}$ concentration gradient across the lysosomal membrane is established and maintained is poorly understood. The most well understood $Ca^{2+}$ store in the cell is the ER. Upon store depletion, the luminal sensor protein STIM1 oligomerizes to activate the highly $Ca^{2+}$-selective ORAI/CRAC channels on the plasma membrane, refilling the ER $Ca^{2+}$ store via the SERCA pump (*Clapham, 2007*; *Lewis, 2007*; *Berridge, 2012*). However, depletion of lysosomal $Ca^{2+}$ stores

does not induce extracellular $Ca^{2+}$ entry (*Haller et al., 1996b*). The endocytic pathway may theoretically deliver extracellular $Ca^{2+}$ to lysosomes. However, most $Ca^{2+}$ taken up through endocytosis is lost quickly during the initial course of endosomal acidification prior to reaching lysosomes during endosome maturation (*Gerasimenko et al., 1998*). In various cell types, when the lysosomal pH gradient is dissipated, either by inhibiting the V-ATPase or by alkalizing reagents such as $NH_4Cl$, free *luminal* $[Ca^{2+}]_{Ly}$ was found to drop drastically (*Calcraft et al., 2009*; *Christensen et al., 2002*; *Dickson et al., 2012*; *Lloyd-Evans et al., 2008*; *Shen et al., 2012*), with no or very small concomitant increase in *cytosolic* $Ca^{2+}$ (*Christensen et al., 2002*; *Dickson et al., 2012*). These findings have been interpreted to mean that the proton gradient in the lysosome is responsible for actively driving $Ca^{2+}$ into the lysosome via an unidentified $H^+$-dependent $Ca^{2+}$ transporter (*Morgan et al., 2011*). Because these findings are consistent with studies in yeast showing that the $Ca^{2+}/H^+$ exchangers establish the vacuolar $Ca^{2+}$ gradient (*Morgan et al., 2011*), this 'pH hypothesis' has been widely accepted (*Calcraft et al., 2009*; *Christensen et al., 2002*; *Lloyd-Evans et al., 2008*; *Morgan et al., 2011*; *Shen et al., 2012*). However, large, prolonged manipulations of luminal pH may interfere directly with $Ca^{2+}$ reporters, and secondarily affect many other lysosomal processes, especially lysosome luminal $Ca^{2+}$ buffering (*Dickson et al., 2012*), lysosome membrane potential, and fusion/fission of endosomes and lysosomes (*Mindell, 2012*). Therefore, these hypotheses about lysosomal $Ca^{2+}$ refilling and store maintenance remain to be tested under more physiological conditions. Directly measuring lysosomal $Ca^{2+}$ release has been made possible recently by using lysosome-targeted genetically-encoded $Ca^{2+}$ indicators (*Shen et al., 2012*) (GCaMP3-ML1; see *Figure 1—figure supplement 1A*), which co-localized well, in healthy cells, with lysosomal associated membrane protein-1 (Lamp1), but not with markers for the ER, mitochondria, or early endosomes (*Figure 1—figure supplement 1B*).

## Results

### A physiological assay to monitor lysosomal $Ca^{2+}$ refilling

Monitoring lysosomal $Ca^{2+}$ store refilling requires direct activation of lysosomal $Ca^{2+}$ channels with specific agonists to repeatedly induce $Ca^{2+}$ release. NAADP, the only known endogenous $Ca^{2+}$-mobilizing messenger that has been suggested to be lysosome-specific, was not useful due to its membrane impermeability and strong desensitization (*Morgan et al., 2011*). Using the specific, membrane–permeable synthetic agonists that we recently identified for lysosomal TRPML1 channels (ML-SA1) (*Shen et al., 2012*), we developed a lysosomal $Ca^{2+}$ refilling assay as shown in *Figure 1A*. In HEK293 cell lines stably-expressing GCaMP3-ML1 (HEK-GCaMP3-ML1 cells), bath application of ML-SA1 (30s) in a 'zero' (low; free $[Ca^{2+}]$ <10 nM) $Ca^{2+}$ external solution produced robust lysosomal $Ca^{2+}$ release measured by GCaMP3 fluorescence ($\triangle F/F_0$ >0.5; *Figure 1A,B*, *Figure 1—figure supplement 1*, *Figure 1—figure supplement 2A*). The membrane-permeable form of the fast $Ca^{2+}$ chelator BAPTA (BAPTA-AM) completely blocked the ML-SA1 response (*Figure 1—figure supplement 1D*), supporting its $Ca^{2+}$ specificity. Importantly, GCaMP3-ML1-tagged lysosomes co-localized well with LysoTracker, indicating that the pH of these lysosomes was not different from lysosomes without GCaMP3-ML1 (*Figure 1—figure supplement 1E*).

After release of the initial, 'naïve' $Ca^{2+}$ store upon first application of ML-SA1, lysosomal $Ca^{2+}$ stores are largely depleted, as immediate re-application of ML-SA1 evoked much smaller or no response (*Figure 1—figure supplement 2B*). The reduction in the second response was unlikely caused by channel desensitization, as surface-expressed TRPML1 mutant (TRPML1-4A [*Shen et al., 2012*]) showed repeated $Ca^{2+}$ entry in $Ca^{2+}$-containing (2 mM) external solution (*Figure 1—figure supplement 2C*). Notably, increasing the time interval between consecutive applications quickly and effectively restored the lysosomal ML-SA1 responses; it required 5 min for full restoration/refilling (*Figure 1—figure supplement 2D–F*). With 5 min of refilling time (chosen for the rest of our experiments), in healthy HEK-GCaMP3-ML1 cells, the second and third ML-SA1 responses are often slightly higher than the first, naïve response (*Figure 1A,B*).

To ensure the ML-SA1-induced $Ca^{2+}$ responses are exclusively intracellular and lysosomal, all ML-SA1 responses were measured either in the 'zero' $Ca^{2+}$ external solution (*Figure 1A*) or in the presence of $La^{3+}$ (*Figure 1—figure supplement 2G,H*), a membrane-impermeable TRPML channel blocker (*Dong et al., 2008*) that is expected to completely inhibit surface-expressed TRPML1

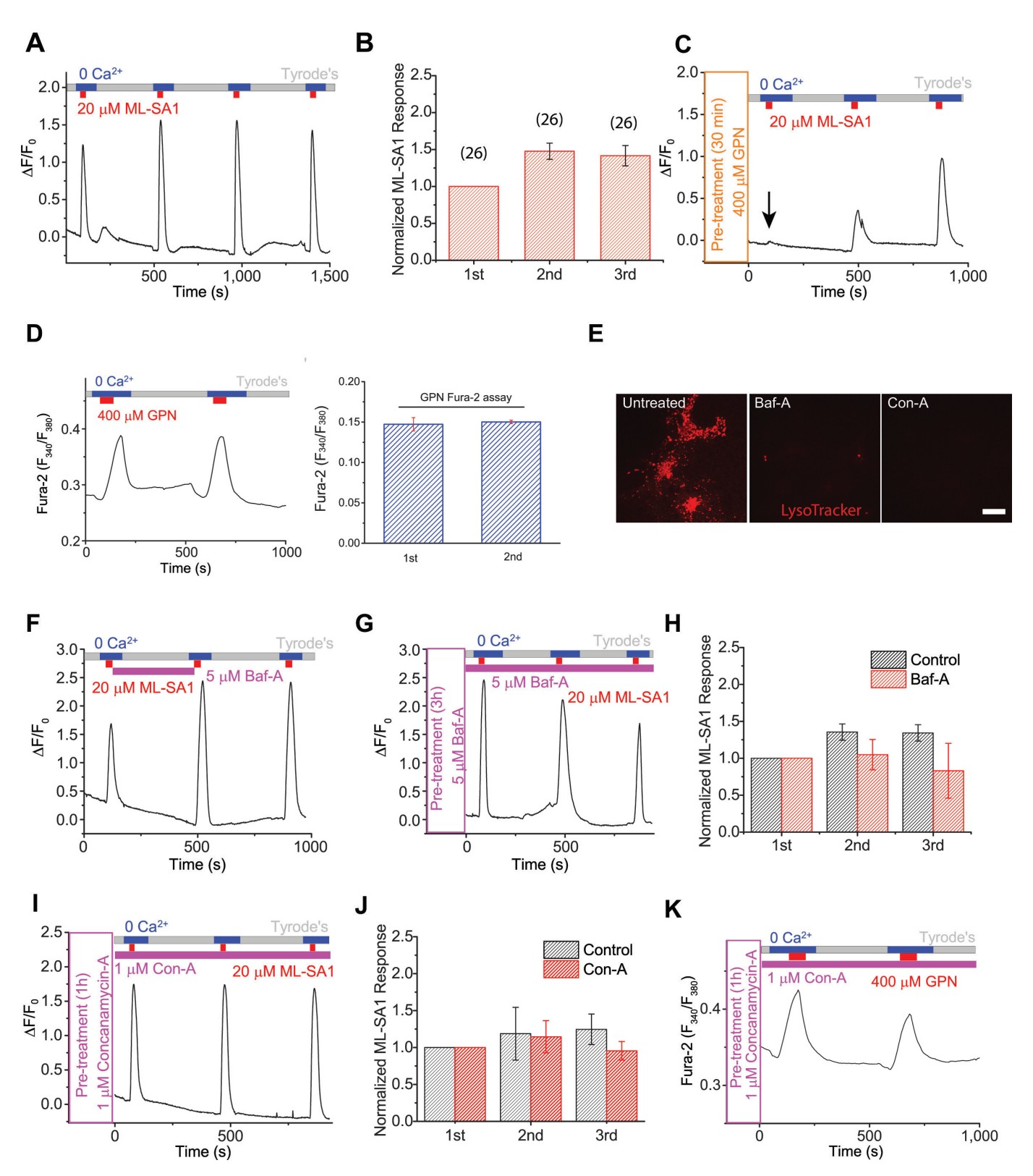

**Figure 1.** The proton gradient of the lysosome is not required for lysosomal $Ca^{2+}$ store refilling. (**A**) In HEK293 cells stably expressing GCaMP3-ML1 (HEK-GCaMP3-ML1 cells), bath application of the ML1 channel agonist ML-SA1 (20 µM) in a low or 'zero' $Ca^{2+}$ (free $[Ca^{2+}]$<10 nM) external solution induced an increase in GCaMP3 fluorescence ($F_{470}$). After washout for 5 min, repeated applications of ML-SA1 induced responses that were similar to or

*Figure 1 continued on next page*

*Figure 1 continued*

larger than the first one. Note that because baseline may drift during the entire course of the experiment (up to 20 min), we typically set $F_0$ based on the value that is closest to the baseline. (B) The average $Ca^{2+}$ responses of three ML-SA1 applications at intervals of 5 min (n=26 coverslips; *Figure 1—source data 1*). (C) Pre-treatment with lysosome-disrupting agent GPN for 30 min abolished the response to ML-SA1 in HEK-GCaMP3-ML1 cells. Washout of GPN resulted in a gradual re-appearance of ML-SA1 responses. See quantitation in *Figure 1—figure supplement 2K*. (D) Repeated applications of GPN resulted in $Ca^{2+}$ release that was measured with the $Ca^{2+}$-sensitive dye Fura-2 ($F_{340}/F_{380}$) in non-transfected HEK293T cells. (E) Application of Bafilomycin-A (Baf-A, 5 μM) and Concanamycin-A (Con-A, 1 μM) quickly (<5 min) abolished LysoTracker staining, an indicator of acidic compartments. (F) Acute application of Baf-A (5 μM) for 5 min did not block $Ca^{2+}$ refilling of lysosomes in HEK-GCaMP3-ML1 cells. (G) Prolonged pre-treatment (3 hr) with Baf-A did not block $Ca^{2+}$ refilling of lysosomes. (H) Quantification of 1st (p value = 0.11), 2nd (p=0.01), and 3rd (p=0.004) ML-SA1 responses upon Baf-A treatment (n=8) compared to control traces (n=6; *Figure 1—source data 1*). (I) Prolonged treatment (1 hr) with Con-A did not prevent lysosomes from refilling their $Ca^{2+}$ stores. (J) Quantification of 1st (p=0.90), 2nd (p=0.33), and 3rd (p=0.66) ML-SA1 responses with Con-A pre-treatment (n=3; *Figure 1—source data 1*). (K) Con-A did not reveal differences in $Ca^{2+}$ refilling responses to repeated applications of GPN in untransfected HEK293T cells. Panels A, C, D, F, G, I, and K are the average of 30–40 cells from one representative coverslip/experiment. The data in panels B, H, and J represent mean ± SEM from at least three independent experiments.

The following source data and figure supplements are available for figure 1:

**Source data 1.** Source data of *Figure 1B, H, J*: The average $Ca^{2+}$ responses to ML-SA1 applications under control (B), Baf-A1 treatment (H), and Con-A treatment (J).

**Figure supplement 1.** A lysosome-targeted genetically-encoded $Ca^{2+}$ indicator to measure lysosomal $Ca^{2+}$ release, store depletion, and refilling.

**Figure supplement 2.** An assay to monitor lysosomal $Ca^{2+}$ store depletion and refilling.

**Figure supplement 3.** GPN and ML-SA1 have different effects on lysosome pH and GCaMP3 fluorescence.

**Figure supplement 4.** Inhibition of PI(3,5)P$_2$ production does not prevent lysosomal $Ca^{2+}$ refilling.

channels. $Ca^{2+}$ release was completely blocked by the TRPML-specific, synthetic antagonists ML-SI1 or ML-SI3 (*Figure 1—figure supplement 2I,J*). In addition, pretreatment with the lysosome-disrupting reagent Glycyl-L-phenylalanine 2-naphthylamide (GPN) (*Berg et al., 1994*) also completely abolished the refilling either in 'zero' $Ca^{2+}$ or in the presence of $La^{3+}$ (*Figure 1C*; *Figure 1—figure supplement 2G*), further supporting the lysosome-specificity of the response. The effect of GPN, presumably on so-called 'lysosomal membrane permeabilization', is known to be rapid and reversible (i.e. membrane 'resealing') (*Kilpatrick et al., 2013*). Consistently, washout of GPN led to gradual recovery of ML-SA1 responses (*Figure 1C*; *Figure 1—figure supplement 2K*). Similar $Ca^{2+}$ refilling of lysosomes was also observed in GCaMP3-ML1-transfected human fibroblasts (*Figure 1—figure supplement 2L*), Cos-7 cells (*Figure 1—figure supplement 2M*), primary mouse macrophages, mouse myoblasts, and DT40 chicken B cells (*Figure 3D–E'*). These findings support the idea that these responses are mediated by intracellular $Ca^{2+}$ release from refilled lysosomal stores (also see *Figure 1—figure supplement 1C* for signals from individual lysosomes). Taken together, these results ensure that lysosomal $Ca^{2+}$ stores can be emptied and refilled repeatedly and consistently in a time-dependent manner.

## Studying lysosomal $Ca^{2+}$ refilling using a lysosome-specific 'membrane-permeabilizer'

GPN is a membrane-permeable di-peptide that is broken down by the lysosome-specific enzyme Cathepsin C. GPN causes permeabilization of lysosome membranes resulting from the accumulation of its breakdown products within lysosomes (*Berg et al., 1994*). Because it is a lysosome-specific membrane disrupting agent, it is often used to mobilize lysosome-specific $Ca^{2+}$ stores (*Jadot et al., 1984*; *Morgan et al., 2011*; *Berg et al., 1994*; *Haller et al., 1996a*; *Haller et al., 1996b*). Using Fura-2 $Ca^{2+}$ imaging in non-transfected HEK293T cells, repeated applications of GPN resulted in a response of similar magnitude to the first, suggestive of $Ca^{2+}$ refilling (*Figure 1D*). Importantly, in HEK-GCaMP3-ML1 cells, pre-treatment with GPN or BAPTA-AM abolished the initial response to ML-SA1, confirming the GCaMP3-ML1 probe's lysosome and $Ca^{2+}$ specificity (*Figure 1C*).

The GPN-mediated 'membrane permeabilizaton' causes the leakage of small solutes including $Ca^{2+}$ and $H^+$ into the cytosol (*Appelqvist et al., 2012*), resulting in changes in the pH (see *Figure 1—figure supplement 3A*) and $[Ca^{2+}]$ in both the lysosome lumen and the peri-lysosomal (juxta-lysosomal) cytosol (*Berg et al., 1994*; *Kilpatrick et al., 2013*; *Appelqvist et al., 2012*). We therefore tested the $Ca^{2+}$-specificity of GPN-induced increases on the Fura-2 and GCaMP3 signals. In cells pretreated with BAPTA-AM, whereas ER-mediated $Ca^{2+}$ responses were abolished, GPN-induced Fura-2 increases were much reduced but not abolished (*Figure 1—figure supplement 3B,C*). Consistently, in HEK-GCaMP3-ML1 cells pre-treated with BAPTA-AM, GPN still induced a significant increase in GCaMP3 fluorescence. However, in these BAPTA-AM-treated cells, GPN-induced increases in GCaMP3 responses were completely abolished by a pre-treatment of Bafilomycin-A (Baf-A), a specific inhibitor of the V-ATPase (*Morgan et al., 2011*) (*Figure 1—figure supplement 3D*). Given that both $Ca^{2+}$ dyes and GFP-based $Ca^{2+}$ indicators are known to be sensitive to other ionic factors, particularly pH (*Rudolf et al., 2003*), GPN-induced changes in lysosomal and peri-lysosomal pH could directly or indirectly account for the BAPTA-insensitive GCaMP3 and residual Fura-2 signals. Consistent with this prediction, in the vacuoles isolated from HEK-GCaMP3-ML1 cells, GCaMP3 fluorescence was sensitive not only to high $Ca^{2+}$, but also to low pH (*Figure 1—figure supplement 3E*). Because ratiometric dyes are less susceptible to pH changes (*Morgan et al., 2015*), in the Fura-2 assay, GPN may induce a large $Ca^{2+}$ signal, but also a pH-mediated contaminating non-$Ca^{2+}$ signal (compare *Figure 1—figure supplement 3B with C*).

## The pH gradient and V-ATPase are not required for lysosome $Ca^{2+}$ refilling

Next, we investigated the mechanisms underlying $Ca^{2+}$ refilling of lysosomes. Inhibition of endocytosis using dynasore and organelle mobility using cytoskeleton inhibitors such as nocodazole and trichostatin A did not block refilling (data not shown). Furthermore, disruption of Golgi function using Brefeldin-A also had no effect on refilling (*Figure 2—figure supplement 1A*). Hence, in agreement with previous findings, the secretory and endocytic pathways are not directly involved in $Ca^{2+}$ refilling. PI(3,5)P$_2$ is a lysosome-specific phosphoinositide that regulates multiple lysosomal channels and transporters including ML1 (*Xu and Ren, 2015*). Pharmacologically decreasing PI(3,5)P$_2$ levels using two small molecule PIKfyve inhibitors: YM201636 (*Jefferies et al., 2008*) and Apilimod (*Xu et al., 2013*) did not prevent lysosomal $Ca^{2+}$ refilling (*Figure 1—figure supplement 4A,B*).

Previous findings have suggested that the pH gradient in the lysosome may be important to $Ca^{2+}$ refilling (*Xu and Ren, 2015*; *Morgan et al., 2011*), however few studies have carefully investigated this possibility. Baf-A and Concanamycin-A (Con-A), inhibitors of the V-ATPase, increase the pH of the lysosome (*Morgan et al., 2011*), demonstrated by abolishing LysoTracker staining within minutes after application (*Figure 1E*). Surprisingly, acute application of Baf-A did not affect the response to ML-SA1, and had little effect on refilling (*Figure 1F*), nor did pretreatment of Baf-A for 1, 3 (*Figure 1G,H*), or 16 hr. Similarly, pretreatment with Con-A also had no effect on $Ca^{2+}$ refilling of lysosomes (*Figure 1I, J, K*). These findings suggest that contradictory to previous conclusions, the pH gradient and V-ATPase may not be required for $Ca^{2+}$ refilling, and that an alternative mechanism is responsible for supplying $Ca^{2+}$ to lysosomes.

## The Endoplasmic Reticulum (ER) $Ca^{2+}$ store is required for lysosomal $Ca^{2+}$ refilling

Lysosomal $Ca^{2+}$ refilling was drastically reduced upon removal of extracellular $Ca^{2+}$ during refilling time in HEK-GCaMP3-ML1 cells (*Figure 2A*). However, blocking $Ca^{2+}$ entry using the generic cation channel blocker $La^{3+}$ did not prevent refilling (*Figure 2—figure supplement 1B*). Because ER stores are passively, although slowly, depleted in 0 $Ca^{2+}$ (*Wu et al., 2006*) (also see *Figure 2—figure supplement 1C*), given the demonstrated role of extracellular $Ca^{2+}$ in ER store refilling (*Lewis, 2007*; *Berridge, 2012*), we investigated the role of the ER in lysosomal refilling. Thapsigargin (TG), a specific inhibitor of the ER SERCA pump (*Thastrup et al., 1990*), rapidly and completely abolished $Ca^{2+}$ refilling to lysosomes (*Figure 2B,G*), but did not affect the first, naïve ML-SA1 response (*Figure 2C*; second response marked with arrow) or lysosomal pH (*Figure 2D*). In the GPN & Fura-2 assay that provides a reasonable (but not perfect; see above) measurement of lysosomal $Ca^{2+}$ release independent of ML1, TG application also largely reduced the second GPN response (*Figure 1D, 2E*), which

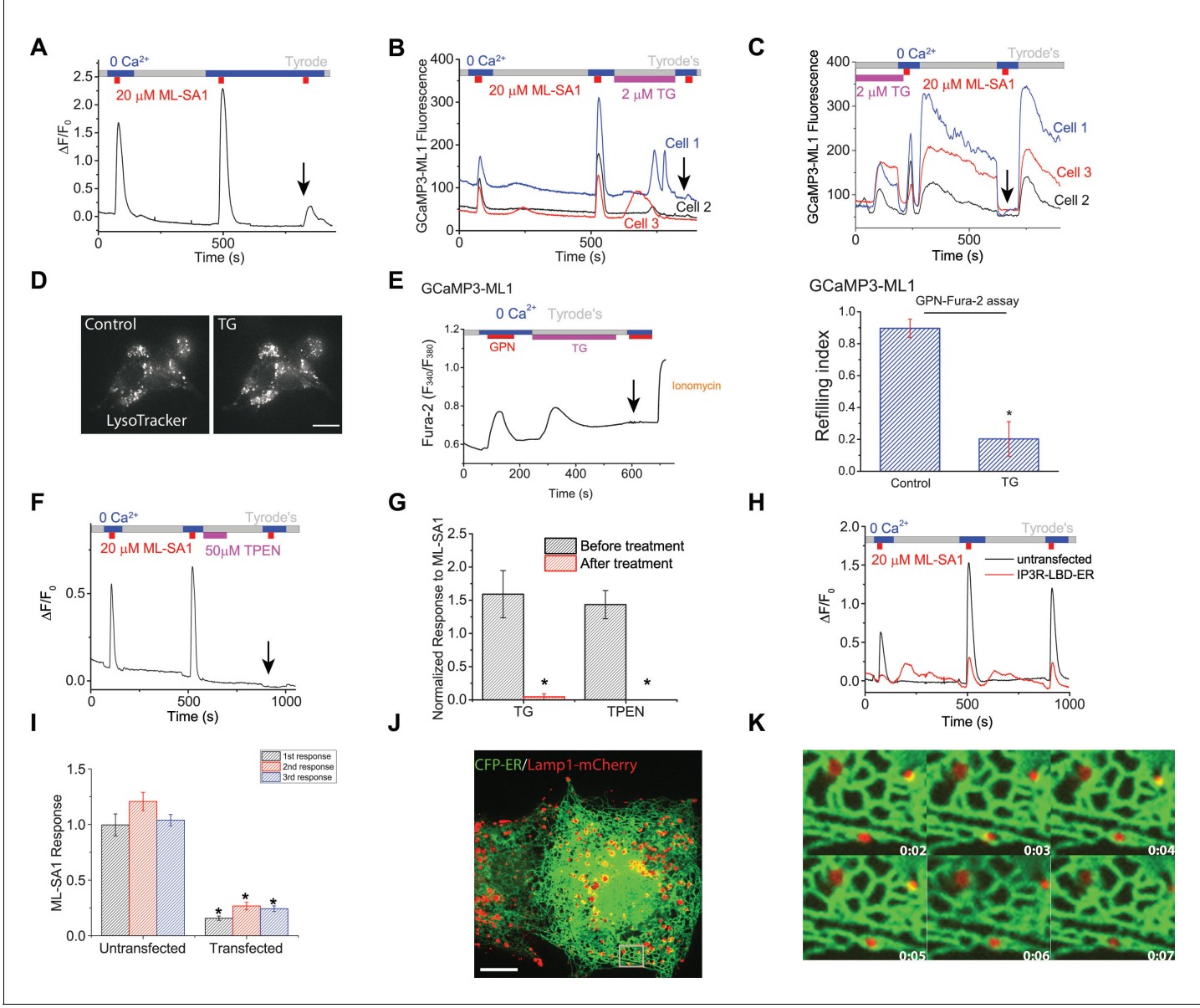

**Figure 2.** Lysosomal Ca$^{2+}$ refilling is dependent on the endoplasmic reticulum (ER) Ca$^{2+}$. (**A**) Ca$^{2+}$ refilling of lysosomes requires external Ca$^{2+}$. (**B**) Dissipating the ER Ca$^{2+}$ gradient using SERCA pump inhibitor Thapsigargin (TG) blocked lysosomal Ca$^{2+}$ refilling in HEK-GCaMP3-ML1 cells. Three representative cells from among 30–40 cells on one coverslip are shown. Note that Ca$^{2+}$ release from the ER through passive leak revealed after blocking SERCA pumps was readily seen in HEK-GCaMP3-ML1 cells, presumably due to the close proximity between lysosomes and the ER (*Kilpatrick et al., 2013*). (**C**) The effect of acute application of TG (2 µM) on the naïve ML-SA1 response and lysosomal Ca$^{2+}$ refilling in HEK-GCaMP3-ML1 cells. Application of TG did not affect the naïve, initial response to ML-SA1, but did abolish the refilled response (see arrow). Control naïve response 1.39 ± 0.09 (n=3); Naïve response after TG 1.08 ± 0.07 (n=3); p=0.2024. (**D**) LysoTracker staining was not reduced by TG (2 µM). (**E**) Representative Ca$^{2+}$ imaging trace and statistical data (right panel; *Figure 2—source data 1*) show that TG application reduced the second responses to GPN compared to the control shown in *Figure 1D*. (**F**) Chelating ER Ca$^{2+}$ using 2-min TPEN treatment blocked Ca$^{2+}$ refilling of lysosomes. (**G**) TG (p=0.008; n=5) and TPEN (p=0.001; n=5) abolished Ca$^{2+}$ refilling of lysosomes (*Figure 2—source data 1*). (**H**) In HEK-GCaMP3-ML1 cells that were transiently transfected with the IP3R-ligand binding domain with ER targeting sequence (IP3R-LBD-ER), which significantly reduces basal [Ca$^{2+}$]$_{ER}$ (see *Figure 2—figure supplement 2E*), ML-SA1 responses were reduced, compared to untransfected cells on the same coverslip. (**I**) The 1st (p=0.0014), 2nd (p=0.0004), and 3rd responses (p<0.0001) of GCaMP3-ML1 cells transfected with the IP3R-LBD-ER were significantly reduced compared to untransfected cells on the same coverslip (n=5; *Figure 2—source data 1*). (**J**) Lysosomes (labeled with Lamp1-mCherry) interact closely with the ER (labeled with CFP-ER). (**K**) Time lapse zoomed-in images of a selected region from **J** show the dynamics of ER-lysosome association (see an example in the boxed area). Panels **A**, **F**, **H** are the average responses of 30–40 cells from one representative experiment. The data in panel **G** represent mean ± SEM from five independent experiments.

*Figure 2 continued on next page*

*Figure 2 continued*

The following source data and figure supplements are available for figure 2:

**Source data 1.** Comparisons of GPN (E) and ML-SA1 responses (G, I) under different pharmacological and genetic manipulations (*Figure 2E,G,I*).
**Figure supplement 1.** The ER $Ca^{2+}$ store regulates lysosome $Ca^{2+}$ stores.
**Figure supplement 2.** The ER $Ca^{2+}$ store regulates lysosome $Ca^{2+}$ stores.

could be further reduced or abolished by Baf-A pretreatment. These results suggest that TG had no direct effect on the naïve $Ca^{2+}$ store in lysosomes, but specifically and potently affected lysosomal $Ca^{2+}$ refilling. A rapid and complete block of $Ca^{2+}$ refilling was also observed for another SERCA pump inhibitor CPA (*Figure 2—figure supplement 1D–G*). TG may induce an unfolded protein response (UPR; *Matsumoto et al., 2013*). However, the UPR inducer Tunicamycin (*Oslowski and Urano, 2011*) did not affect refilling (*Figure 2—figure supplement 1H,I*).

$[Ca^{2+}]_{ER}$, but not cytosolic $Ca^{2+}$, can be chelated by *N,N,N',N'-Tetrakis* (2-pyridylmethyl) ethylenediamine (TPEN), a membrane-permeable metal chelator with a low affinity for $Ca^{2+}$ (*Hofer et al., 1998*). Although TPEN may also enter the lysosomal lumen, the much reduced (>100 fold less) $Ca^{2+}$-binding affinity in the acidic pH ($pH_{LY} = 4.6$) suggests that chelation of lysosomal $Ca^{2+}$ would be minimal. Acute application of TPEN completely blocked lysosomal $Ca^{2+}$ refilling (*Figure 2F,G*). A short application of TPEN also blocked ER $Ca^{2+}$ release stimulated by the endogenous P2Y receptor agonist ATP (*Figure 2—figure supplement 2B* compared to *Figure 2—figure supplement 2A*), but not the lysosomal Fura-2 $Ca^{2+}$ response stimulated by GPN (*Figure 2—figure supplement 2C* compared to *Figure 2—figure supplement 2A*). These findings suggest that chelation of ER $Ca^{2+}$ stores using TPEN had no direct effect on the naïve $Ca^{2+}$ store in lysosomes, but specifically and potently affected lysosomal $Ca^{2+}$ refilling.

The ER $Ca^{2+}$ store can also be genetically and chronically reduced without raising intracellular $Ca^{2+}$ levels by transfecting cells with the IP3R ligand-binding domain with an ER targeting sequence (IP3R-LBD-ER) (*Várnai et al., 2005*). As expected, IP3R-LBD-ER expression decreased ATP-induced IP3R-mediated $Ca^{2+}$ release (*Figure 2—figure supplement 2E*). Interestingly, it also reduced the GPN induced lysosomal $Ca^{2+}$ release in HEK293T cells (*Figure 2—figure supplement 2E*). Furthermore, in HEK-GCaMP3-ML1 cells transfected with IP3R-LBD-ER, lysosomal $Ca^{2+}$ release was significantly reduced when compared to un-transfected cells on the same coverslip (*Figure 2H,I*). Collectively, these findings suggest that the ER, the major $Ca^{2+}$ store in the cell, is essential for refilling and the ongoing maintenance of lysosomal $Ca^{2+}$ stores, but not required for the naïve $Ca^{2+}$ release from lysosomes.

A functional interaction between ER and lysosome $Ca^{2+}$ stores was previously suggested (*Haller et al., 1996a*; *1996b*), but these results have been largely ignored, presumably due to the lack of specific tools required for definitive interpretation. Recent findings have shown that as endosomes mature, they increase their contact with the ER (*Friedman et al., 2013*). Interestingly, the $Ca^{2+}$ released from SERCA inhibition on the ER was detected on our GCaMP3-ML1 probe (*Figure 2B,C*, *Figure 2—figure supplement 1D,F*), likely due to close membrane contact between the ER and lysosomes (*Eden, 2016*). Similar detection of ER $Ca^{2+}$ release by a genetically-encoded, lysosomally-targeted chameleon $Ca^{2+}$ sensor utilizing lysosome membrane protein Lamp1 has also been reported (*McCue et al., 2013*). Using time-lapse confocal imaging, we found that the majority of lysosomes, marked by Lamp1-mCherry, move and traffic together with ER tubules, labeled with CFP-ER (*Figure 2J,K*). Thus, the ER could be the direct source of $Ca^{2+}$ to lysosomes by forming nanojunctions with them (*Eden, 2016*).

## IP3-receptors, not ryanodine receptors, on the ER are required for $Ca^{2+}$ refilling of lysosomes

$Ca^{2+}$ release from the ER is mediated primarily by two $Ca^{2+}$ release channels, IP3Rs and ryanodine receptors (RYRs), both of which are expressed in HEK cells (*Querfurth et al., 1998*; *Jurkovicova et al., 2008*) (see also *Figure 2—figure supplement 2F*). Since IP3Rs are responsible for $Ca^{2+}$ transfer to mitochondria (*Hayashi et al., 2009*), we examined whether IP3Rs on the ER

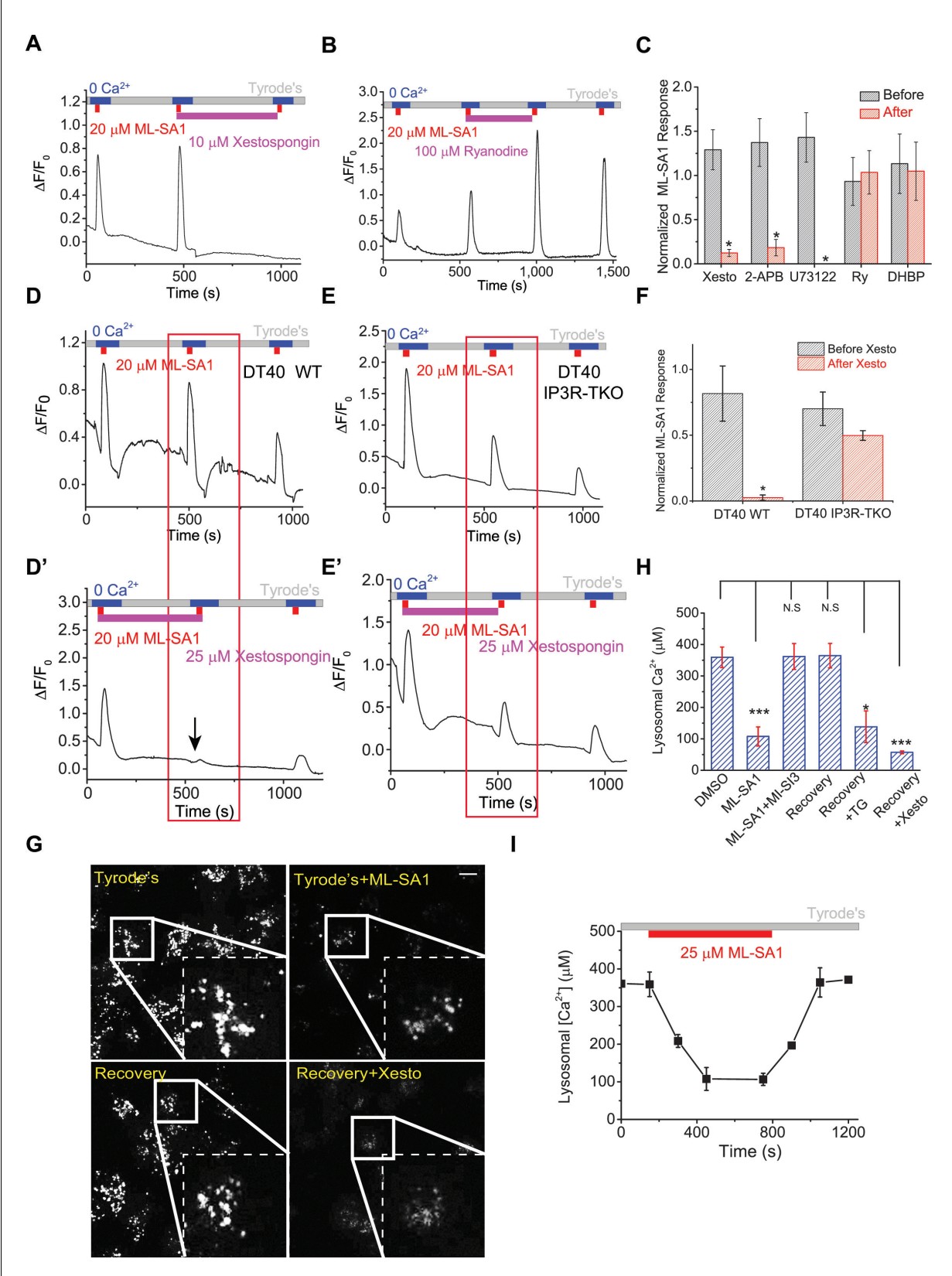

**Figure 3.** IP3-receptors on the ER are required for lysosomal Ca²⁺ store refilling. (**A**) The IP3-receptor (IP3R) antagonist Xestospongin-C (Xesto, 10 µM) prevented Ca²⁺ refilling of lysosomes in HEK-GCaMP3-ML1 cells (p=0.007). Note that Xesto was co-applied with ML-SA1. (**B**) Ryanodine (100 µM), which

*Figure 3 continued on next page*

Figure 3 continued

blocks Ryanodine receptors at high concentrations, did not block Ca²⁺ refilling to lysosomes. Note that Ryanodine was co-applied with ML-SA1. (C) Quantification of the responses to ML-SA1 in HEK-GCaMP3-ML1 cells after treatment with Xesto, 2-APB (*Figure 3—figure supplement 1K*), U73122 (*Figure 3—figure supplement 1L,M*), Ryanodine (Ry), and DHBP (*Figure 3—figure supplement 2A*) (*Figure 3—source data 1*).(D) DT40 WT cells transiently transfected with GCaMP3-ML1 show Ca²⁺ refilling. (D') IP3R antagonist Xesto completely blocked Ca²⁺ refilling of lysosomes in DT40 WT cells. (E) DT40 IP3R triple KO (TKO) cells transiently transfected with GCaMP3-ML1 also show Ca²⁺ refilling. (E') Xesto did not block Ca²⁺ refilling of lysosomes in IP3R-TKO cells. (F) Quantification of ML-SA1 responses with or without Xesto in WT and IP3R-TKO DT40 cells (*Figure 3—source data 1*). (G) Representative images showing the effects of Xesto on the recovery of ML-SA1-induced responses in HEK-ML1 stable cells loaded with OG-BAPTA-dextran. La³⁺ was used to block external Ca²⁺ influx that could be mediated by surface-expressed ML1 in the overexpression system (see *Figure 1—figure supplement 2G*). (H) The effects of TG and Xesto on intralysosomal Ca²⁺ contents measured by OG-BAPTA-dextran (*Figure 3—source data 1*). (I) The effects of ML-SA1 on [Ca²⁺]$_{Ly}$ measured by OG-BAPTA-dextran. Panels A, B, D, D', E, E', F, F' and H are the average of 30–40 cells from one representative experiment. The data in panels C, F and H represent mean ± SEM from at five independent experiments. The scale bar in panel G = 10 µm.

The following source data and figure supplements are available for figure 3:

**Source data 1.** Normalized ML-SA1 responses or lysosomal Ca²⁺ contents under pharmacological (C, H) or genetic manipulations (F) (*Figure 3C, F, H*).
**Figure supplement 1.** ER IP3-Receptors regulate Ca²⁺ refilling of lysosomes.
**Figure supplement 2.** Lysosomal Ca2+ refilling is compromised in IP3R TKO DT40 cells.
**Figure supplement 3.** Measuring lysosomal Ca²⁺ release with lysosome-targeted luminal Ca²⁺ indicators.

were responsible for Ca²⁺ refilling of the lysosome. Notably, Ca²⁺ refilling of the non-naïve lysosome Ca²⁺ store was completely blocked by Xestospongin-C (Xesto; *Figure 3A,C*), a relatively specific IP3R blocker (*Peppiatt et al., 2003*) (*Figure 3—figure supplement 1A–E*). In addition, in the GPN & Fura-2 assay that provides a measurement of lysosomal Ca²⁺ release independent of ML1, blocking IP3 receptor by Xesto profoundly attenuated lysosomal Ca²⁺ refilling in both HEK-GCaMP3-ML1 cells and non-transfected mouse embryonic fibroblasts (MEF) cells (*Figure 1D*; *Figure 3—figure supplement 1F–J*). Acute application of Xesto after allowing lysosomal Ca²⁺ stores to refill for 5 min (hence stores are completely refilled and functionally equivalent to 'naïve' ones) also slowly (up to 10 min) reduced lysosomal Ca²⁺ release, suggesting that constitutive lysosomal Ca²⁺ release under resting conditions may gradually deplete lysosome Ca²⁺ stores if refilling is prevented (*Figure 3—figure supplement 1B–E*). Consistent with this hypothesis, long-term (20 min) treatment with the aforementioned ER Ca²⁺ manipulators including TG and TPEN almost completely abolished lyso-somal Ca²⁺ release (*Figure 2—figure supplement 2D*), further supporting the interpretation that ongoing constitutive Ca²⁺ release and refilling requires ER Ca²⁺.

2-APB, a non-specific IP3R antagonist (*Peppiatt et al., 2003*), also blocked Ca²⁺ refilling (*Figure 3C*, *Figure 3—figure supplement 1K*). U73122 is a PLC inhibitor that blocks the constitutive production of IP3 (*Cárdenas et al., 2010*) and prevents ATP-induced IP3R-mediated Ca²⁺ release (*Figure 3—figure supplement 1L*). U73122 also completely prevented Ca²⁺ refilling of lysosomes (*Figure 3C*, *Figure 3—figure supplement 1M*), suggesting that basal production of IP3 is essential for Ca²⁺ refilling of lysosomes. In contrast, blocking the RyRs with high (>10 µM) concentrations of ryanodine (*Figure 3B,C*), or with the receptor antagonist 1,1'-diheptyl-4,4'-bipyridinium (DHBP) (*Berridge, 2012*) (*Figure 3C*, *Figure 3—figure supplement 2A*), did not affect Ca²⁺ refilling. Notably, co-application of RYR and IP3R blockers with the second ML-SA1 response did not change the amplitude of the response (*Figure 3A,B*). Together, these findings demonstrate that IP3Rs on the ER are specifically required for lysosomal Ca²⁺ refilling, but not for Ca²⁺ release from naïve stores or completely refilled stores.

In contrast with the pharmacological analyses described above, lysosomal store refilling occurred in both WT and IP3R triple KO (TKO) DT40 chicken B cells (*Várnai et al., 2005*; *Cárdenas et al., 2010*) that were transfected with GCaMP3-ML1 (*Figure 3D–F*, *Figure 3—figure supplement 2B*). However, unlike WT DT40 cells, in which the IP3R-specific antagonist Xesto completely blocked Ca²⁺ refilling (*Figure 3D', F*), Xesto had no obvious blocking effect in IP3R-TKO cells (*Figure 3E', F*). In addition, the kinetics of lysosomal refilling was markedly delayed in IP3R TKO cells compared with

WT cells (*Figure 3—figure supplement 2C*). These results are consistent with the notion that in normal conditions, IP3Rs are the sole source of $Ca^{2+}$ refilling of lysosomes. When IP3Rs are genetically deleted, however, IP3R-independent mechanisms contribute to lysosomal $Ca^{2+}$ refilling, possibly as a consequence of genetic compensation. Refilling in IP3R-TKO DT40 cells was not blocked by RYR inhibitors (*Figure 3—figure supplement 2D,E*).

## Studying lysosomal $Ca^{2+}$ refilling using intra-lysosomal $Ca^{2+}$ dyes

As an additional assay to directly 'monitor' intralysosomal $Ca^{2+}$, we employed two intraluminal $Ca^{2+}$ indicators Fura-Dextran and Oregon 488 BAPTA-1 dextran (OG-BAPTA-dextran) (*Morgan et al., 2015*). After being pulse/chased into ML1-mCherry-transfected HEK293T cells or HEK-ML1 stable cells, the dyes enter the lysosome lumen (*Figure 3—figure supplement 3A,B*) after endocytosis (*Christensen et al., 2002*). Due to their pH sensitivities, these dyes can detect intra-lysosomal $Ca^{2+}$ ($[Ca^{2+}]_{LY}$) changes, but preferentially only when the intra-lysosomal pH ($pH_L$) remains constant below pH 5.0 (*Morgan et al., 2015*) (see *Figure 3—figure supplement 3C*). In the Fura-Dextran-loaded ML1-mCherry-transfected HEK-293T cells, ML-SA1 application induced $Ca^{2+}$ release from the lysosome lumen (*Figure 3—figure supplement 3D*). As we found in our GCaMP3-ML1 assay, Xesto abolished the ML-SA1-induced decrease in $[Ca^{2+}]_{LY}$ (*Figure 3—figure supplement 3D,E*). Likewise, in HEK-ML1 stable cells loaded with OG-BAPTA-dextran, which had a much higher efficiency in loading to the lysosome (*Figure 3—figure supplement 3B*), TG or Xesto treatment profoundly reduced lysosomal $Ca^{2+}$ refilling (*Figure 3G—I*; *Figure 3—figure supplement 3F,G*). Note that LysoTracker staining was not significantly reduced by ML-SA1, TG, or Xesto, suggesting that the signals were primarily mediated by changes of intralysosomal $Ca^{2+}$, not intralysosomal pH. In contrast, treatment of cells with Baf-A1 or $NH_4Cl$ markedly increased lysosomal pH from 4.8 to 7.0 (*Figure 3—figure supplement 3J,K*). Such large pH elevations may cause dramatic changes in both $K_d$ of OG-BAPTA-dextran (see *Figure 3—figure supplement 3C*) and luminal $Ca^{2+}$ buffering capability (*Morgan et al., 2015*; *Dickson et al., 2012*), preventing accurate determinations of $[Ca^{2+}]_{LY}$ under these pH manipulations. Taken together, these results are consistent with the conclusions that were drawn based on the aforementioned ML-SA1 & GCaMP3-ML1 assay and the GPN & Fura-2 assay.

## Inhibition of ER IP3R channels and $Ca^{2+}$ release causes lysosomal dysfunction and a LSD-like phenotype

Lysosomal $Ca^{2+}$ is important to lysosomal function and membrane trafficking (*Kiselyov et al., 2010*; *Shen et al., 2012*; *Lloyd-Evans et al., 2008*). Lysosomal dysfunction is commonly associated with a compensatory increase of lysosome biogenesis, manifested as increased expression of essential lysosomal genes (*Settembre et al., 2013*). For example, the expression of Lamp1, a lysosomal marker, is elevated in most LSDs (*Meikle et al., 1997*). Lamp1 expression was significantly elevated in cells treated with low concentrations of IP3R blockers 2-APB and Xesto, as well as the ER $Ca^{2+}$ chelator TPEN, but not in the cells treated with the RyR blocker DHBP (*Figure 4A*). Consistently, LysoTracker staining was significantly increased in cells treated with Xesto, but not 1,1'-diheptyl-4,4'-bipyridinium dibromide (DHBP; *Figure 4B*). Lysosomal dysfunction is also often associated with lysosomal enlargement and accumulation of various incompletely digested biomaterials (*Shen et al., 2012*; *Dong et al., 2008*). Notably, in cells that were treated with Xesto, but not with the vehicle control DMSO, lysosomal compartments were enlarged, and non-degradable, autofluorescent lipofuscin-like materials accumulated in puncta structures (*Figure 4C*), reminiscent of cells with defective lysosomal $Ca^{2+}$ release (ML1 KO cells; *Dong et al., 2008*) (*Figure 4C*). By showing that inhibiting IP3R-mediated $Ca^{2+}$ release from the ER results in a lysosome storage phenotype in the cell, these findings suggest that lysosome $Ca^{2+}$ store refilling from IP3Rs ion the ER has important consequences for lysosome function and cellular health.

## Discussion

Using pharmacological and genetic approaches to manipulate ER $Ca^{2+}$ levels and $Ca^{2+}$ release and three different assays to directly measure lysosome $Ca^{2+}$ release, we show that under normal conditions lysosome $Ca^{2+}$ stores are refilled from the ER $Ca^{2+}$ store through IP3 receptors independent of lysosome pH (see *Figure 4D*). Our findings are in contrast to several studies in the literature that suggest that inhibition of the V-ATPase is sufficient to deplete lysosome $Ca^{2+}$ stores. Previous

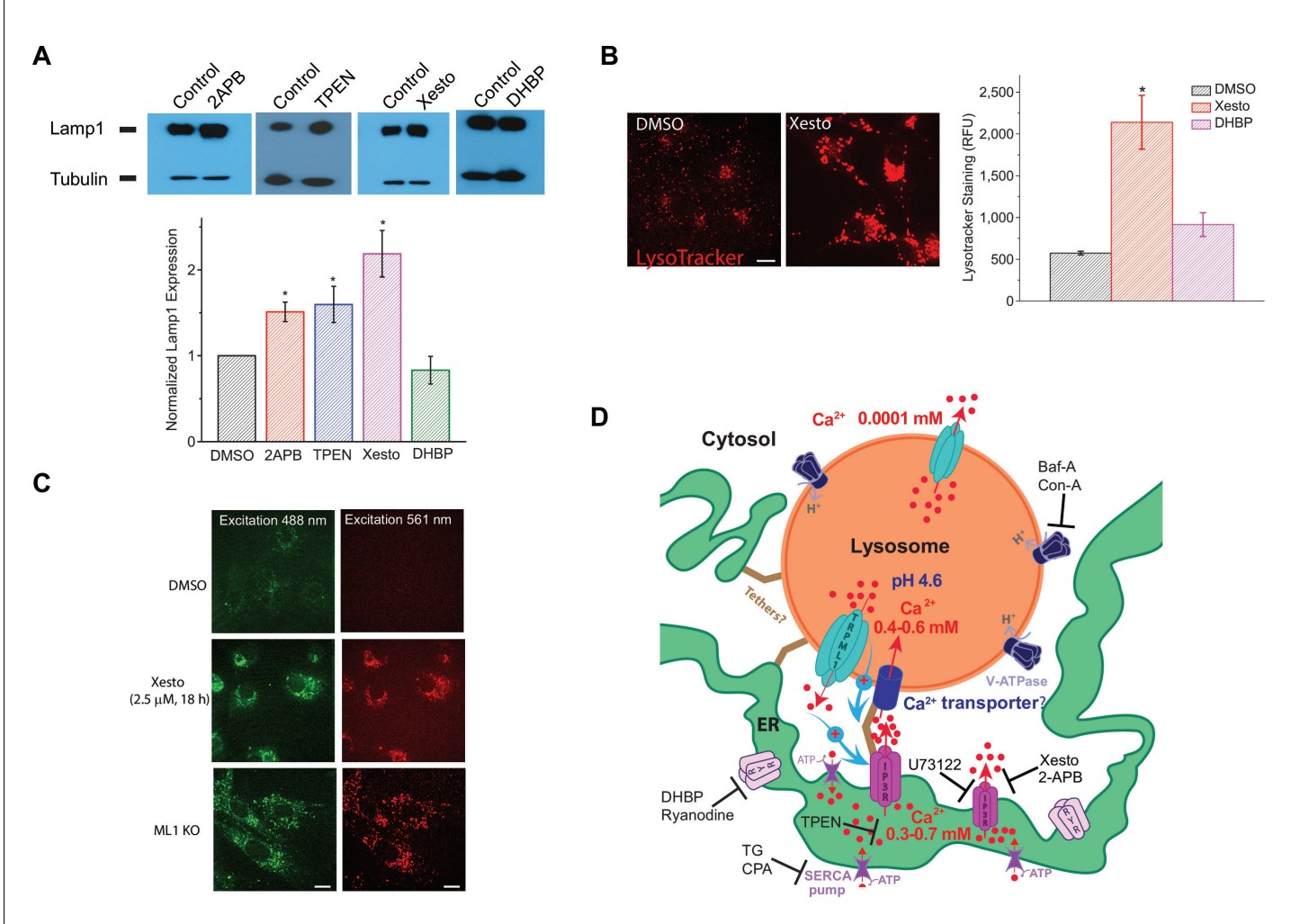

**Figure 4.** Blocking ER IP3-receptors Ca²⁺ channels refill lysosome Ca²⁺ stores to prevent lysosomal dysfunction. (**A**) Upper panels: Western blotting analyses of Lamp1 in HEK293T cells treated with 2-APB (50 µM), TPEN (0.1 µM), Xesto (10 µM), and DHBP (5 µM) compared to DMSO for 24 hr (n=4 separate experiments for each condition). Lower panel: treating HEK293T cells with 2-APB (p=0.05) and Xesto (p=0.013), as well as TPEN (p=0.02), significantly increased Lamp1 expression. DHBP did not significantly change Lamp1 expression (p=0.23) (*Figure 4—source data 1*). (**B**) The effects of Xesto (10 µM, 18 hr; p=0.0001) and DHBP (50 µM, 18 hr; p=0.063) treatment compared to DMSO on the lysosomal compartments detected by LysoTracker staining in HEK293T cells (average of 20–30 cells in each of 3 experiments; *Figure 4—source data 1*). Scale bar = 15 µm. (**C**) The effect of Xesto (10 µM, 18 h) treatment on accumulation of the autofluorescent lipofuscin materials in non-transfected HEK293T cells. Autofluorescence was observed in a wide spectrum but shown at two excitation wavelengths (488 and 561 nm). ML1 KO MEFs are shown for comparison. Scale bar = 15 µm. (**D**) A proposed model of Ca²⁺ transfer from the ER to lysosomes. The ER is a Ca²⁺ store with [Ca²⁺]$_{ER}$ ~ 0.3–0.7 mM; lysosomes are acidic (pH$_{Ly}$ ~ 4.6) Ca²⁺ stores ([Ca²⁺]$_{Ly}$ ~ 0.5 mM). IP3Rs on the ER release Ca²⁺ to produce a local high Ca²⁺ concentration, from which an unknown low-affinity Ca²⁺ transport mechanism refills Ca²⁺ to a lysosome. Unidentified tether proteins may link the ER membrane proteins directly with the lysosomal membrane proteins to maintain contact sites of 20–30 nm for purposes of Ca²⁺ exchange. Ca²⁺ released from lysosomes or a reduction/depletion in [Ca²⁺]$_{Ly}$ may, through unidentified mechanisms, 'promote' or 'stabilize' ER-lysosome interaction (*Phillips and Voeltz, 2016*; *Eden, 2016*). At the functional ER-lysosome contact sites, Ca²⁺ can be transferred from the ER to lysosomes through a passive Ca²⁺ transporter or channel based on the large chemical gradient of Ca²⁺ that is created when lysosome stores are depleted. Baf-A and Con-A are specific V-ATPase inhibitors; Xesto and 2APB are IP3R blockers; U73122 is a PLC inhibitor that blocks the constitutive production of IP3; DHBP and Ryanodine (>10 µM) are specific RyR blockers; TG and CPA are SERCA pump inhibitors; and TPEN is a luminal Ca²⁺ chelator.

The following source data is available for figure 4:

**Source data 1.** Source data of *Figure 4A,B*: Quantifications of Lamp-1 protein levels (**A**) or LysoTracker staining (**B**) under different experimental conditions and manipulations.

conclusions suggesting the importance of H⁺ gradient in regulating lysosome $Ca^{2+}$ stores would therefore implicate the existence of an H⁺-dependent $Ca^{2+}$ transporter in lysosomal membranes that can operate at the extremely low cytosolic free $Ca^{2+}$ level (100 nM), representing a high affinity uptake system. Our work, however, suggest that a low affinity uptake mechanism is more likely. Hence either a low affinity $Ca^{2+}$ transporter or rectifying $Ca^{2+}$ channel might suffice. A putative VDAC-like channel in the lysosome, resembling mitochondrial VDAC channels (*van der Kant and Neefjes, 2014*), may interact directly with IP3Rs to receive $Ca^{2+}$ from the ER. Importantly, it has been previously suggested that $Ca^{2+}$ uptake into isolated lysosomes is mediated by a low-affinity (mM range) $Ca^{2+}$ transporter (*Lemons and Thoene, 1991*).

The lysosomal pH gradient is thought to be essential for the maintenance of high free $[Ca^{2+}]_{Ly}$ (*Calcraft et al., 2009*; *Christensen et al., 2002*; *Dickson et al., 2012*; *Lloyd-Evans et al., 2008*; *Shen et al., 2012*). However, in addition to triggering lysosomal $Ca^{2+}$ release, as proposed by Christensen et al. (*Christensen et al., 2002*), lysosomal pH elevation is also known to affect $[Ca^{2+}]_{Ly}$ or its measurement via several other mechanisms. Whereas the total $[Ca^{2+}]_{Ly}$ is reported to be in the low mM range (5–10 mM), free $[Ca^{2+}]_{Ly}$ is generally agreed to be in the high µM range (100–500 µM) (*Morgan et al., 2015*). Therefore, the lysosome lumen must contain substantial $Ca^{2+}$ buffers (*Morgan et al., 2015*). $Ca^{2+}$ buffers in acidic compartments and the ER are known to bind $Ca^{2+}$ much better at neutral pH (*Dickson et al., 2012*). Hence increasing $pH_L$ from 4.8 to 7.0 may effectively reduce free $[Ca^{2+}]_{Ly}$ without necessarily triggering lysosomal $Ca^{2+}$ release and affecting total $[Ca^{2+}]_{Ly}$. Consistent with such an interpretation, a compelling study recently demonstrated that in secretory granules and the ER, increasing luminal pH changed the $Ca^{2+}$ buffering capacity of both $Ca^{2+}$ containing compartments and reduced free $[Ca^{2+}]$, while causing a minimal (20 nM) increase in cytosolic $Ca^{2+}$ (*Dickson et al., 2012*). Additionally, lysosomal pH may act on luminal $Ca^{2+}$ dyes by affecting their chromophore fluorescence and $Ca^{2+}$-binding affinity ($K_d$) (*Morgan et al., 2015*). Because $K_d$ drops more than 1,000 foldtimes when $pH_L$ is increased from 4.8 to 7.0, accurate calibration is currently not possible. Furthermore, prolonged lysosomal pH manipulations may also indirectly affect lysosomal $Ca^{2+}$ homeostasis, for instance, via membrane fusion and fission between compartments containing different amounts of $Ca^{2+}$, H⁺, and their buffers. Finally, although elevating lysosomal pH may trigger lysosomal $Ca^{2+}$ release, the accompanying increase in cytoplasmic $Ca^{2+}$ was rather small (20–40 nM) (*Dickson et al., 2012*; *Christensen et al., 2002*). Moreover, the instantaneous changes (following pH increase and decrease) of $Ca^{2+}$ probe fluorescence (*Dickson et al., 2012*; *Christensen et al., 2002*) are inconsistent with the slow rates of $Ca^{2+}$ leak and re-uptake demonstrated in the current study.

The persistence of a GPN signal even after intracellular $Ca^{2+}$ chelation is important for understanding the limits of this lysosome-specific pharmacological tool. GPN can certainly be used in conjunction with other tools to examine lysosome specificity, but caution is necessary with its use for $Ca^{2+}$ store measurement, as a component of the signal observed in Fura-2 loaded cells, although small, is a result of the membrane permeabilization that causes a decrease in cytosolic pH. Similarly, reagents like Baf-A and NAADP that are used to mobilize lysosomal $Ca^{2+}$ also release H⁺ into the cytosol (*Morgan and Galione, 2007*; *Appelqvist et al., 2012*; *Scott and Gruenberg, 2011*; *Yoshimori et al., 1991*), which could have been misinterpreted as a $Ca^{2+}$ signal in previous studies (*Morgan et al., 2011*). pH may affect cytosolic $Ca^{2+}$ indicators through the chromophore fluorescence, $Ca^{2+}$-binding affinity, or $Ca^{2+}$-dependent conformational changes (e.g., in the case of GCaMP) (*Morgan et al., 2015*). Therefore, if experimental conditions are not optimized, the presumed cytosolic $Ca^{2+}$ signals may also reflect pH changes, or unidentified pH-mediated non-$Ca^{2+}$ signals. We propose that BAPTA-AM control experiments be routinely conducted in any lysosomal $Ca^{2+}$ measurement. It is possible that the 'pH contaminating effect' might have resulted in numerous misinterpretations of lysosome $Ca^{2+}$ stores in the literature, particularly those examining the interactions between ER and lysosome $Ca^{2+}$.

Based on our results in the current study, recent studies of ER-lysosome interactions (*Phillips and Voeltz, 2016*), and previous $Ca^{2+}$ uptake studies on isolated lysosomes (*Lemons and Thoene, 1991*), we hypothesize that ER-refilling of lysosomal stores is a regulated, two-step process (see *Figure 4D*). First, lysosome store depletion may trigger establishment of ER-lysosome contacts (*Phillips and Voeltz, 2016*). Although lysosomes and ER are in close proximity under resting conditions, lysosome store depletion may 'stabilize' the ER-lysosome contact, and/or 'tether' and approximate both membranes (e.g., from 20–30 nm to 10 nm) (*Phillips and Voeltz, 2016*;

*Eden, 2016*). Second, at the relatively stable, functional ER-lysosome contact sites, a passive $Ca^{2+}$ transport process can occur from the ER to lysosomes, by utilizing the large $Ca^{2+}$ gradient created when lysosome stores are actively depleted. Up to 5 min may be required to complete the whole refilling process from 'initiation' through 'uptake'.

Our results not only provide an explanation for the reported sensitivity of the $Ca^{2+}$ stores of acidic organelles to ER disrupting agents (*Menteyne et al., 2006*; *Haller et al., 1996a*), but are also consistent with the observations that lysosomes may buffer cytosolic $Ca^{2+}$ released from the ER (*López-Sanjurjo et al., 2013*). The unexpected role of the ER in maintaining $Ca^{2+}$ stores in lysosomes may help resolve the long-standing mystery of how impaired ER $Ca^{2+}$ homeostasis is commonly seen in lysosomal storage diseases (LSDs) (*Coen et al., 2012*), and how manipulating ER $Ca^{2+}$ reduces lysosome storage (*Lloyd-Evans et al., 2008*; *Mu et al., 2008*). In addition, our work reveals that, depending on the treatment conditions (acute *versus* prolonged treatment), many assumed-to-be ER-specific reagents may indirectly affect lysosome $Ca^{2+}$ stores. This may impact the interpretations of a large body of literature on $Ca^{2+}$ signaling. Although we demonstrated a central role of IP3Rs in lysosomal $Ca^{2+}$ refilling, other ER $Ca^{2+}$ channels may also participate under certain conditions, as seen in the IP3R TKO cells.

Accumulating evidence suggests that the ER forms membrane contact sites with other organelles, including plasma membrane, mitochondria (*Cárdenas et al., 2010*), endosomes (*Alpy et al., 2013*), and lysosomes (*van der Kant and Neefjes, 2014*). ER-endosome membrane contact, although currently difficult to study, was proposed to facilitate cholesterol transport from endosomes to the ER (*Rocha et al., 2009*; *van der Kant and Neefjes, 2014*; *Drin et al., 2016*). Given the established role of lysosomal $Ca^{2+}$ release in cholesterol transport (*Shen et al., 2012*), lysosomal $Ca^{2+}$ release may have a direct role in regulating ER-lysosome interaction (see *Figure 4D*). In ER-mitochondria contact sites, the tethering protein GRP-75 links IP3Rs with VDAC channels on mitochondria to regulate $Ca^{2+}$ homeostasis and ATP production (*Cárdenas et al., 2010*). Similar unidentified tethers may also link IP3Rs with the putative lysosomal $Ca^{2+}$ transporter for store refilling (see *Figure 4D*). The importance of lysosomal $Ca^{2+}$ in regulating a variety of intracellular signaling pathways is becoming increasingly recognized (*Medina et al., 2015*). ER-lysosome interaction may serve as a hub for $Ca^{2+}$ signaling to regulate cellular homeostasis through coordinating the primary anabolic and catabolic pathways in the cell. Studying the two-step lysosomal $Ca^{2+}$ refilling process may prove important for future identification of the low-affinity $Ca^{2+}$ uptake transporter/channel in the lysosome, and for studying the molecular mechanisms that regulate the functional ER-lysosome interaction.

## Materials and methods

### Molecular biology

Genetically-encoded $Ca^{2+}$ indicator GCaMP3 was fused directly to the N-terminus of ML1 (GCaMP3-ML1) as described previously (*Shen et al., 2012*). The IP3R-LBD-ER construct (*Várnai et al., 2005*) was a kind gift from Dr. Thomas Balla (National Institute of Child Health and Human Development, NIH). The pECFP-ER plasmid was obtained from CLONTECH. Lamp1-mCherry was made by fusing mCherry with the C terminus of Lamp1.

### Western blotting

Standard Western blotting protocols were used. HEK293T cells were treated every 4 hr for 24 hr with IP3R antagonists 2-APB and Xestospongin-C, ER $Ca^{2+}$ chelator TPEN, and RyR antagonist DHBP. Lamp1 antibody was from Developmental Studies Hybridoma Bank (Iowa).

### Mammalian cell culture

Immortalized cell lines (HEK293 and Cos-7) were purchased from ATCC and cultured following standard culture protocols. DT40-WT and IP3R-TKO cells were a generous gift from Dr. Darren Boehning (The University of Texas Health Sciences Center at Houston). Human fibroblasts were obtained from the Cornell Institute for Medical Research (NJ, USA). HEK 293 cells stably expressing GCaMP3-ML1 (HEK-GCaMP3-ML1 cells) were generated using the Flip-In T-Rex 293 cell line (Invitrogen). All these cells were neither authenticated nor tested for mycoplasma contamination. HEK293 cells are on the list of frequently misidentified or cross-contaminated cell lines. All cells were cultured in a 37°C

incubator with 5% $CO_2$. HEK293T cells, Tet-On HEK293 cells stably expressing GCaMP3-ML1 (HEK-GCaMP3-ML1 cells), Cos-7 cells, and human fibroblasts were cultured in DMEM F12 (Invitrogen) supplemented with 10% (vol/vol) FBS or Tet-free FBS. DT40 cells were kept in suspension in RPMI 1640 (Invitrogen) supplemented with 450 µL β-mercaptoethanol, 2 mM L-glutamine, 10% FBS, and 1% chicken serum (*Várnai et al., 2005*; *Cárdenas et al., 2010*). We noted that lysosomal $Ca^{2+}$ store refilling was often compromised in high-passage or poorly-maintained cell cultures.

Human fibroblasts and DT40 cells were transiently transfected using the Invitrogen Neon electroporation kit (1200 V, 1 pulse, 30 s). HEK293T cells, HEK-GCaMP3-ML1 cells, and Cos-7 cells were transfected using Lipofectamine 2000 (Invitrogen). All cells were used for experiments 24–48 hr after transfection.

## Confocal imaging

Live imaging of cells was performed on a heated and humidified stage using a Spinning Disc Confocal Imaging System. The system includes an Olympus IX81 inverted microscope, a 100X Oil objective NA 1.49 (Olympus, UAPON100XOTIRF), a CSU-X1 scanner (Yokogawa), an iXon EM-CCD camera (Andor). MetaMorph Advanced Imaging acquisition software v.7.7.8.0 (Molecular Devices) was used to acquire and analyze all images. LysoTracker (50 nM; Invitrogen) was dissolved in culture medium and loaded into cells for 30 min before imaging. MitoTracker was dissolved in culture medium and loaded into cells for 15 min before imaging (25 nM). Coverslips were washed 3 times with Tyrode's and imaged in Tyrode's.

## GCaMP3-ML1 $Ca^{2+}$ imaging

GCaMP3-ML1 expression was induced in Tet-On HEK-GCaMP3-ML1 cells 20-24h prior to experiments using 0.0 1µg/mL doxycycline. GCaMP3-ML1 fluorescence was monitored at an excitation wavelength of 470 nm ($F_{470}$) using a EasyRatio Pro system (PTI). Cells were bathed in Tyrode's solution containing 145 mM NaCl, 5 mM KCl, 2 mM $CaCl_2$, 1 mM $MgCl_2$, 10 mM Glucose, and 20 mM Hepes (pH 7.4). Lysosomal $Ca^{2+}$ release was measured in a zero $Ca^{2+}$ solution containing 145 mM NaCl, 5 mM KCl, 3 mM $MgCl_2$, 10 mM glucose, 1 mM EGTA, and 20 mM HEPES (pH 7.4). $Ca^{2+}$ concentration in the nominally free $Ca^{2+}$ solution is estimated to be 1–10 µM. With 1 mM EGTA, the free $Ca^{2+}$ concentration is estimated to be <10 nM based on the Maxchelator software (http://maxchelator.stanford.edu/). Experiments were carried out 0.5 to 6 hr after plating. Because baseline may drift during the entire course of the experiment (up to 20 min), we typically set $F_0$ based on the value that is closest to the baseline.

## Fura-2 $Ca^{2+}$ imaging

Cells were loaded with Fura-2 (3 µM) and Plurionic-F127 (3 µM) in the culture medium at 37°C for 60 min. Florescence was recorded using the EasyRatio Pro system (PTI) at two different wavelengths (340 and 380 nm) and the ratio ($F_{340}/F_{380}$) was used to calculate changes in intracellular $[Ca^{2+}]$. All experiments were carried out 1.5 to 6 hr after plating.

## Oregon green 488 BAPTA-1 dextran imaging

Cells were loaded with Oregon Green 488 BAPTA-1 dextran (100 µg/ml) at 37°C in the culture medium for 4–12 hr, and then pulsed/chased for additional 4–16 hr. Fluorescence imaging was performed at 37°C. In vitro calcium-binding ($K_d$) affinities of OG-BAPTA-dextran were determined using KCl-based solutions (140 mM KCl, X mM $CaCl_2$, 1 mM $MgCl_2$, 10 mM HEPES, 10 mM MES, 0 or 1 mM BAPTA) adjusted to different pH (pH 4.5, 5.0, 6.0, and 7.0). By varying the amount of added $Ca^{2+}$ (X= 0- 10 mM), solutions with different pH and free $[Ca^{2+}]$ were made based on the Maxchelator software (http://maxchelator.stanford.edu/). OG-BAPTA-dextran (5 µg/ml) fluorescence for each solution was obtained to plot the calibration curve (*Morgan et al., 2015*; *Dickson et al., 2012*; *Christensen et al., 2002*). In cells that were pre-treated with ionomycin, nigericin, and valinomycin (*Morgan et al., 2015*; *Dickson et al., 2012*; *Christensen et al., 2002*), in vivo minimal and maximal Fluorescence ($F_{min}$ and $F_{max}$) were determined by perfusing the cells with 0 or 10 mM $Ca^{2+}$ external solutions, respectively. Lysosomal $[Ca^{2+}]$ at different pH were determined using the following calibration equation: $[Ca^{2+}] = K_d \times (F-F_{min})/ (F_{max}-F)$.

## Lysosomal pH measurement

Cells were pulsed with OG-488-dextran for 6 hr, and chased for additional 12 hr (*Johnson et al., 2016*). Cells were then bathed in the external solutions (145 mM KCl, 5 mM glucose, 1 mM MgCl$_2$, 1 mM CaCl$_2$, 10 mM HEPES, 10 mM MES, adjusted to various pH values ranging from 4.0 to 8.0) that contained 10 µM nigericin and 10 µM monensin (*Johnson et al., 2016*). Images were captured using an EasyRatio Pro system. A pH standard curve was plotted based on the fluorescence ratios: $F_{480}/F_{430}$.

## Cytosolic pH sensitivity of GCaMP3-ML1

GCaMP3-ML1-positive vacuoles were isolated from vacuolin-1-treated HEK-GCaMP3-ML1 cells, as described previously (*Zhang et al., 2012*). Briefly, cells were treated with 1 µM vacuolin-1 for up to 12 hr to increase the size of late endosomes and lysosomes (*Cerny et al., 2004*). Vacuoles were released into the dish by mechanical disruption of the cell membrane with a small glass electrode. After vacuoles were released into the dish, patch pipettes containing either a 'high-Ca$^{2+}$' (10 mM) internal solution or a 'low-pH' solution (140 mM KCl, 1 mM EGTA, 20 mM MES, 10 mM Glucose, pH adjusted to 2.0) were placed close to 'puff' the vacuoles. Images were captured using a CCD camera connected to the fluorescence microscope.

## Reagents

All reagents were dissolved and stored in DMSO or water and then diluted in Tyrode's and 0 Ca$^{2+}$ solutions for experiments. 2-APB, ATP, Con-A, CPA, Doxycycline, DHBP, TG, TPEN were from Sigma; GPN and U73122 were from Santa Cruz; Ryanodine was from Abcam; LysoTracker, Fura-2, Mitotracker, Plurionic F-127, and Fura-Dextran were from Invitrogen; Baf-A was from LC Laboratories; ML-SA1 was from Chembridge; and Xestospongin-C was from Cayman Chemical, AG Scientific, and Enzo; Oregon Green 488 BAPTA-1 dextran was from life technologies. ML-SI compounds were identified from a Ca$^{2+}$-imaging-based highthroughput screening conducted at NIH/NCATS Chemical Genomics Center (NCGC;https://pubchem.ncbi.nlm.nih.gov/bioassay/624414#section=Top). ML-SI compounds are available upon request.

## Data analysis

Data are presented as mean ± SEM. All statistical analyses were conducted using GraphPad Prism. Paired t-tests were used to compare the average of three or more experiments between treated and untreated conditions. A value of $p < 0.05$ was considered statistically significant. In the cases only individual traces were shown, the traces are representative from at least 30–40 cells, or from at least independent repeats.

## Acknowledgement

We thank Dr. Darren Boehning for DT40-WT and IP3R-TKO cells, and Thomas Balla for the IP3R-LBD-ER construct. We also thank Richard Hume and Edward Stuenkel for comments on the manuscript, and appreciate the encouragement and helpful comments of other lab members in the Xu lab.

## Additional information

### Funding

| Funder | Grant reference number | Author |
| --- | --- | --- |
| NIH Office of the Director | NS062792 | Abigail G Garrity<br>Wuyang Wang<br>Crystal MD Collier<br>Sara A Levey<br>Qiong Gao<br>Haoxing Xu |

| NIH Office of the Director | AR060837 | Abigail G Garrity |
| | | Wuyang Wang |
| | | Crystal MD Collier |
| | | Sara A Levey |
| | | Qiong Gao |
| | | Haoxing Xu |

The funders had no role in study design, data collection and interpretation, or the decision to submit the work for publication.

## Author contributions

AGG, WW, Conception and design, Acquisition of data, Analysis and interpretation of data, Drafting or revising the article, Contributed unpublished essential data or reagents; CMDC, SAL, QG, Acquisition of data, Analysis and interpretation of data, Contributed unpublished essential data or reagents; HX, Conception and design, Analysis and interpretation of data, Drafting or revising the article, Contributed unpublished essential data or reagents

## Author ORCIDs

Haoxing Xu, http://orcid.org/0000-0003-3561-4654

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
