## [Decision Letter]

Thank you for submitting your article "The Endoplasmic Reticulum, Not the pH Gradient in the Lysosome, Are the Source of Calcium to Lysosomes" for consideration by *eLife*. Your article has been reviewed by three peer reviewers, and the evaluation has been overseen by David Clapham as the Reviewing Editor and Eve Marder as the Senior Editor. One of the three reviewers, Murali Prakriya, has agreed to share his identity.

The reviewers have discussed the reviews with one another and the Reviewing Editor has drafted this decision to help you prepare a revised submission.

Summary:

Garrity et al. challenge the previously held view that lysosomal V-ATPase driven-acidification provides the driving force for lysosomal Ca^2+^ refilling. Using various manipulations designed to inhibit the proton pump (Baf-A or Con-A) and several Ca^2+^ imaging tools that directly and indirectly monitor lysosomal Ca^2+^, the authors show that the proton pump by is likely not responsible for refilling of the lysosomal Ca^2+^ store. Rather the authors report that manipulations that deplete ER Ca^2+^ stores also affect refilling of the lysosomal compartment, suggesting a functional and possibly physical coupling of the ER and lysosomal Ca^2+^ stores. The authors include new results using OG-BAPTA-dextran showing that depletion of ER Ca^2+^ stores by thapsigargin, or inhibition of IP3Rs by the membrane permeable Xestospongin inhibited the lysosomal Ca^2+^ content. An interesting point is that the previously postulated H^+^-dependent Ca^2+^ transporter must operate with cytosolic free Ca^2+^ level of 100 nM and therefore be extremely high affinity. The present mechanism, although not identified, would be a low affinity uptake mechanism.

Overall the data represent quite a challenging and intriguing new concept. Although most of the data appear convincing, some major open questions remain to be addressed. We feel these can be remedied within a month by additional experiments.

Essential revisions:

1) You need at least one convincing experiment showing that inhibiting the V-ATPase increases pH (needs to be measured) but doesn't affect store [Ca^2+^]. In the Christensen studies, inhibiting V-ATPase with Baf (similar to the treatment used in this paper) increased luminal pH to ~ 7.0 (where the Ca^2+^ imaging dye should work well) and drastically decreased [Ca^2+^]lys from ~ mM to μM. Although you added new OG-BAPTA imaging, it needs calibration. You should show the data measuring the pH and [Ca^2+^]. The bar graphs (Figure 3 and Figure 3—figure supplement 3) do not provide information about the kinetics of the Ca^2+^ change and how they may (or may not) correlate with the Ca^2+^ changes reported by the GCaMP ML1 indicator shown in the majority of the data. Example traces of the lysosomal Ca^2+^ indicator in response to lysosomal and ER Ca^2+^ store depletion should be shown.

2) The authors contend that previous [Ca^2+^]lumen measurements were unreliable because of the pH-sensitivity of the dyes. In the Christensen et al. studies, however, the pH of each lysosome was monitored and used to calibrate the *K*_d_ of the dye. The Christensen studies (using lysosomes that do not overexpress any Ca-releasing channel) suggest that [Ca^2+^]lumen faithfully follows [pH]lumen. Other studies using 45Ca uptake measurement also support the importance of pH gradient (e.g. Lemons & Thoene 1991, JBC 266: 14378). However, the pH sensitivity of the indicators used in the present paper were not tested in situ. We suggest you test the pH sensitivity of your Ca^2+^ indicator in situ by placing a small mouth pipette electrode containing low pH solution+ x [Ca^2+^] next to tagged channel indicator and measure its sensitivity to Ca^2+^ changes. Please provide more detail in the Discussion why you think the previously published measurements were erroneous.

3) A triple IP3R knockout cell line was used to assess the proposed role of IP3Rs in refilling the lysosomal store. However, in contrast to the authors' theory, this does not have a lysosomal Ca^2+^ defect. The authors attribute this to an unknown compensatory mechanism. Please examine whether the Xestospongin abolishes ML1-SA1 responses in the triple IP3R KO cells and show the kinetics of the response. If the responses in the IP3R KO are not affected, and the kinetics now differ, the results would provide an additional layer of confidence for the proposed role of IP3Rs in lysosomal refilling.

---

## [Author Response]

1) You need at least one convincing experiment showing that inhibiting the V-ATPase increases pH (needs to be measured) but doesn't affect store [Ca^2+^]. In the Christensen studies, inhibiting V-ATPase with Baf (similar to the treatment used in this paper) increased luminal pH to ~ 7.0 (where the Ca^2+^ imaging dye should work well) and drastically decreased [Ca^2+^]lys from ~ mM to μM. Although you added new OG-BAPTA imaging, it needs calibration. You should show the data measuring the pH and [Ca^2+^].

We performed the experiments as suggested by the reviewers. First, we monitored lysosomal pH under different conditions using Oregon green 488-dextran, a commonly- used luminal pH indicator [1, 2] (see Methods and new Figure 3—figure supplement 3). Using the calibration that we made for OG-488-dextran (see Figure 3—figure supplement 3), we showed that lysosomal pH (pH_L_), which is ~4.8 under the resting condition, increased dramatically within minutes to ~ pH 7.0 upon treatment of Baf-A1 (5 μM) or NH4Cl (10 mM) in HEK293 cells stably expressing ML1 (see Figure 3—figure supplement 3). Second, based on the calibration for the luminal Ca^2+^ dye OG-BAPTA-dextran (see Figure 3—figure supplement 3), we found that the intra-lysosomal [Ca^2+^] ([Ca^2+^]Ly), which was ~ 360 μM under the resting condition, dropped quickly to ~ 100 μM upon ML-SA1 application (see Figure 3). Note that at pH 4.5, OG-BAPTA- dextran’s *K*d is ~300 μM (Figure 3—figure supplement 3), allowing measurement of Ca^2+^ ranging from 30 (0.1 *K*d) to 3,000 (10 *K*d) μM [3]. At the resting pH_L_ of 4.8 in the HEK cells, OG-BAPTA- dextran’s working range is 130-1,300 μM (*K*d ~ 130 μM). Overall, all these values are in rough agreement with previous measurements [1, 3, 4].

Upon Baf-A1 and NH4Cl treatment to increase lysosomal pH, the fluorescence intensity of OG-BAPTA-dextran increased mildly, following the pH_L_ changes (Figure 3—figure supplement 3). If the effect of lysosomal pH is exclusively on the *K*d of OG-BAPTA-dextran, these results would suggest that [Ca^2+^]Ly dropped significantly, as concluded by Christensen et al. [5]. However, in addition to triggering lysosomal Ca^2+^ release, as proposed by Christensen et al. [5], lysosomal pH elevation is known to affect [Ca^2+^]Ly or its measurement via several additional mechanisms. Therefore, the interpretation offered by Christensen et al. might be too simplistic.

First, a change in lysosomal pH may cause a significant change in luminal Ca^2+^ buffering. In most studies dealing with cytosolic Ca^2+^, the total versus free [Ca^2+^] in the intracellular organelles (stores) are often not separately-considered since these two parameters are interrelated. Whereas the total [Ca^2+^]Ly is reported to be in the low mM range (5-10 mM), free [Ca^2+^]Ly is generally agreed to be in the high μM range (100-500 μM; 360 μM in the current study) [3]. Therefore, the lysosome lumen must contain a substantial amount of Ca^2+^ buffers [3]. Ca^2+^ buffers in the acidic compartments and ER are known to bind Ca^2+^ much better at neutral pH (H^+^ and Ca^2+^ compete with each other for Ca^2+^ buffers) [4]. Hence increasing pH_L_ from 4.8 to 7.0 may effectively reduce free [Ca^2+^]Ly without necessarily triggering lysosomal Ca^2+^ release and affecting total [Ca^2+^]Ly.

Second, lysosomal pH may act on luminal Ca^2+^ dyes by affecting their chromophore fluorescence and Ca^2+^-binding affinity (*K*d) [3]. Because *K*d is dropped more than 1,000 times when pH_L_ is increased from 4.8 to 7.0 (Figure 3—figure supplement 3), perfect calibration is currently impossible. Note that in the Christensen studies, which were conducted in the activated macrophages, the basal pH_L_is ~ pH 3.8 [5]. When pH_L_was increased from 3.8 to 5.0, the free [Ca^2+^]Ly was already dropped to ~ 10 μM (see Figure 3 of ref. [5]). However, in most studies including the current study, lysosomal pH is between 4.5 to 5.0 [1]. Hence, the mechanism proposed by Christensen et al., might not apply to most cell types whose basal pH_L_is close to pH 5.0. In addition, because of high levels of luminal Ca^2+^ buffers, the amount of releasable Ca^2+^ is high. Hence, a depletion of free [Ca^2+^]Ly to low μM is expected to produce a large cytosolic Ca^2+^ increase, which did not occur (see below). In the current study, upon ML-SA1 application to deplete the stores, free [Ca^2+^]Ly dropped from 360 to 100 μM (Figure 3). Hence, within physiological ranges of lysosomal pH (4.8 -7.0) and Ca^2+^, the fluorescence signals of OG-BAPTA- dextran dyes might be nearly saturated, and this would prevent accurate determinations of [Ca^2+^]Ly changes when pH_L_is increased from 4.8 to 7.0 upon Baf-A1 or NH4Cl application. Note that at pH 7.0, OG-BAPTA-dextran’s *K*d is ~0.1 μM (Figure—figure supplement 3C), which is not suitable for measuring Ca^2+^ over 10 μM.

Third, prolonged lysosomal pH manipulations may also indirectly affect lysosomal Ca^2+^ homeostasis, for instance, via membrane fusion and fission between compartments containing different amounts of Ca^2+^, H^+^, and their buffers.

Fourth,although elevating lysosomal pH might trigger lysosomal Ca^2+^ release, the accompanied increase in cytoplasmic Ca^2+^ was rather small (20-40 nM) (see refs. [4, 5]). This minor increase is more consistent with the interpretation that other pH-dependent processes contribute to the changes of OG-BAPTA fluorescence upon Baf-A1 and NH4Cl application. In addition, the instantaneous changes (following pH increase and decrease) of Ca^2+^ probe fluorescence (see Figure 5 in ref. [5] and Figure 3 in ref. [4]) are inconsistent with the slow rates of Ca^2+^ leak (τ = 6 min; see Figure 3—figure supplement 1) and re-uptake (τ = 2 min; see Figure 1—figure supplement 2 & Figure 3—figure supplement 2). Furthermore, given the pH-sensitivity of the Fura-2 dyes, even the observed minor increase (20-40 nM) in cytosolic Ca^2+^, based on instantaneous increases of Fura-2 ratios, could be mostly due to the effects of lysosomal proton release on the Fura-2 dyes (see Figure 1—figure supplement 3).

Because of the reasons listed above, in the current study, we used the luminal Ca^2+^ dyes for free [Ca^2+^]Ly measurement only in the experiments during which the pH remains unchanged.

The bar graphs (Figure 3 and Figure 3—figure supplement 3) do not provide information about the kinetics of the Ca2+ change and how they may (or may not) correlate with the Ca2+ changes reported by the GCamP ML1 indicator shown in the majority of the data. Example traces of the lysosomal Ca2+ indicator in response to lysosomal and ER Ca2+ store depletion should be shown.

The example traces were now added into new Figure 3 and Figure 3—figure supplement 3. Note that due to the photo-bleaching and small signal, we had to minimize the problem by reducing data acquisition. With the calibration curve, we measured lysosomal Ca^2+^ changes under different experimental manipulations. Whereas ML-SA1 application significantly depleted lysosomal Ca^2+^ contents (see new Figure 3), TG treatment only slightly reduced lysosomal Ca^2+^ after 5- 10 min (Figure 3—figure supplement 3). Although a quick drop in [Ca^2+^]Ly may cause a transient increase in cytosolic [Ca^2+^], it may also liberate certain amount of Ca^2+^ from the Ca^2+^-bound buffers, resulting in a delayed reduction in free [Ca^2+^]Ly. Therefore, as suspected by the reviewer, the kinetics of the GCaMP3-ML1 and OG-BAPTA-dextran were not perfectly correlated.

This is an excellent suggestion – thank you! See response #1 regarding [Ca^2+^]Ly measurement.

In GCaMP3-ML1-positive vacuoles that were isolated from HEK-GCaMP3-ML1 cells and released into the dish, we moved patch pipettes containing either a “high Ca^2+^” (10 mM) internal solution or a “low pH” solution (140 mM KCl, 1 mM EGTA, 20 mM MES, 10 mM Glucose, pH adjusted to pH 2.0) close to the vacuoles to produce a “puffing” effect. We showed that GCaMP3- ML1 responded to both Ca^2+^ and pH (see Figure 1—figure supplement 3), but not to the “puffing” of the standard internal solution. These data suggest that our GCaMP3-ML1 probe, in addition to being a good indicator for measuring Ca^2+^ released from lysosomes, may also respond to lysosomal proton release, for instance, upon GPN and Baf-A1 application. Our results suggest that caution is necessary in designing the experiments to study the interaction between lysosomal Ca^2+^ and lysosomal pH. We propose that BAPTA-AM control experiments should be routinely performed in any lysosomal Ca^2+^ studies.

Please provide more detail in the Discussion why you think the previously published measurements were erroneous.

We added a new paragraph in the Discussion (second paragraph), and included a discussion in the Results.

Results: “In contrast, treatment of cells with Baf-A1 or NH_4_Cl markedly increased lysosomal pH from 4.8 to 7.0 (Figure 3—figure supplement 3). Such large pH elevations may cause dramatic changes in both *K*_d_ of OG-BAPTA-dextran (see Figure 3—figure supplement 3) and luminal Ca^2+^ buffering capability (Dickson et al., 2012; Morgan et al., 2015), preventing accurate determinations of [Ca^2+]^LY under these pH manipulations”.

Discussion:“Lysosomal pH gradient is thought to be essential for the maintenance of high free [Ca^2+^]Ly (Calcraft et al., 2009; Christensen et al., 2002; Dickson et al., 2012; Lloyd-Evans et al., 2008; Shen et al., 2012). However, in addition to triggering lysosomal Ca^2+^ release, as proposed by Christensen et al. (Christensen et al., 2002), lysosomal pH elevation is also known to affect [Ca^2+^]Ly or its measurement via several other mechanisms. Whereas the total [Ca^2+^]Ly is reported to be in the low mM range (5-10 mM), free [Ca^2+^]Ly is generally agreed to be in the high μM range (100-500 μM) (Morgan et al., 2015). Therefore, lysosome lumen must contain substantial amount of Ca^2+^ buffers (Morgan et al., 2015). Ca^2+^ buffers in the acidic compartments and ER are known to bind Ca^2+^ much better at neutral pH (Dickson et al., 2012). Hence increasing pH_L_from 4.8 to 7.0 may effectively reduce free [Ca^2+^]Ly without necessarily triggering lysosomal Ca^2+^ release and affecting total [Ca^2+^]Ly. Consistent with such interpretation, a compelling study recently demonstrated that in secretory granules and the ER, increasing luminal pH changed the Ca^2+^ buffering capacity of both Ca^2+^ containing compartments intraluminally to reduce free [Ca^2+^], while causing a minimal (20 nM) increase in cytosolic Ca^2+^ (Dickson et al., 2012). Additionally, lysosomal pH may act on luminal Ca^2+^ dyes by affecting their chromophore fluorescence and Ca^2+^-binding affinity (*K*d) (Morgan et al., 2015). Because *K*d is dropped more than 1,000 times when pH_L_is increased from 4.8 to 7.0, perfect calibration is near impossible. Furthermore, prolonged lysosomal pH manipulations may also indirectly affect lysosomal Ca^2+^ homeostasis, for instance, via membrane fusion and fission between compartments containing different amounts of Ca^2+^, H^+^, and their buffers. Finally, although elevating lysosomal pH may trigger lysosomal Ca^2+^ release, the accompanied increase in cytoplasmic Ca^2+^ was rather small (20- 40 nM) (Christensen et al., 2002; Dickson et al., 2012). Moreover, the instantaneous changes (following pH increase and decrease) of Ca^2+^ probe fluorescence (Christensen et al., 2002; Dickson et al., 2012) are inconsistent with the slow rates of Ca^2+^ leak and re-uptake demonstrated in the current study”.

3) A triple IP3R knockout cell line was used to assess the proposed role of IP3Rs in refilling the lysosomal store. However, in contrast to the authors' theory, this does not have a lysosomal Ca2+ defect. The authors attribute this to an unknown compensatory mechanism. Please examine whether the Xestospongin abolishes ML1-SA1 responses in the triple IP3R KO cells and show the kinetics of the response. If the responses in the IP3R KO are not affected, and the kinetics now differ, the results would provide an additional layer of confidence for the proposed role of IP3Rs in lysosomal refilling.

This is another excellent suggestion. We performed these experiments as suggested, and showed that although lysosomal Ca^2+^ refilling still occurred in IP3R TKO cells, the refilling kinetics were much slower than WT cells (see Figure 3—figure supplement 2), and more importantly, refilling were completely insensitive to IP3R inhibitions (see Figure 3).

References:

1) Johnson, D.E., et al., The position of lysosomes within the cell determines their luminal pH. J Cell Biol, 2016. 212(6): p. 677-92.

2) DiCiccio, J.E. and B.E. Steinberg, Lysosomal pH and analysis of the counter ion pathways that support acidification. J Gen Physiol, 2011. 137(4): p. 385-90.

3) Morgan, A.J., L.C. Davis, and A. Galione, Imaging approaches to measuring lysosomal calcium. Methods Cell Biol, 2015. 126: p. 159-95.

4) Dickson, E.J., et al., Orai-STIM-mediated Ca2+ release from secretory granules revealed by a targeted Ca2+ and pH probe. Proceedings of the National Academy of Sciences of the United States of America, 2012. 109(51): p. E3539-48.

5) Christensen, K.A., J.T. Myers, and J.A. Swanson, pH-dependent regulation of lysosomal calcium in macrophages. Journal of cell science, 2002. 115(Pt 3): p. 599-607.